# Unaccompanied or Separated Migrant Children and Adolescents at the Colombian–Venezuelan Border: Loss of the Social Moratorium and Its Implications

**Carolina Ramírez-Martínez** [1,*] **, Neida Albornoz-Arias** [2] **, Leida Marcela Martínez Becerra** [1] **and Karla Gabriela Tamayo Ramírez** [1]

1   Facultad de Ciencias Jurídicas y Sociales, Universidad Simón Bolívar, Cúcuta 540003, Colombia; marcela_m26@hotmail.com (L.M.M.B.); k_tamayo@unisimon.edu.co (K.G.T.R.)
2   Facultad de Administración y Negocios, Universidad Simón Bolívar, Cúcuta 540003, Colombia; neida.albornoz@unisimon.edu.co
*   Correspondence: carolina.ramirez@unisimon.edu.co

**Abstract:** This study explains the Venezuelan migration involving unaccompanied or separated adolescents (UASA) on the Colombian–Venezuelan border, specifically in Norte de Santander-Táchira. This explanation is framed within the concept of the social moratorium, highlighting three subcategories that contribute to the early abandonment of childhood: 1. the violation of rights, 2. working life, and 3. confrontation of dangers. These subcategories compel UASA to transition prematurely into youth, assuming social, labor, or family responsibilities. Methodologically, we adopt a narrative approach, conducting group interviews with 24 immigrant children and adolescents. Furthermore, 14 interviews are conducted in 2 local markets, and the remaining 10 on 2 central avenues in the city of Cúcuta, Colombia. We conduct a theoretical analysis drawing upon key concepts, including the social moratorium, social constructionism, interaction, and moral and cognitive development. This theoretical framework helps us understand the consequences for the life prospects of this generation. They arise from factors such as school dropout, exposure to health risks, and the absence of free leisure time. These indicators reflect socioeconomic problems, including poverty, abuse, and violence.

**Keywords:** unaccompanied or separated migrant adolescents; social moratorium; denial of rights; prospective reality; violence in migrant environments





## 1. Introduction

Understanding migration as a social and historical phenomenon in the construction of humanity does not diminish its inherent complexity. This complexity becomes particularly pronounced in survival situations where adults grapple with grief, displacement, exposure to acts of violence, and mental health issues (Salaberria Irízar and Sánchez Haro 2017; Achotegui 2012). However, within this landscape, child migration presents itself as a phenomenon that is challenging to measure and address. Some authors have denied its characterization as forced displacement because children and adolescents are relocated from one place to another under the authority of parents or caregivers. In these instances, they become participants in a family migration project that may not correspond with their own interests (Novales Casado 2015; Parella Rubio 2012).

When considering adolescent migration as an evolving field of knowledge, recent advancements have shed light on the specific negative effects on their lives. These effects stem from the challenging conditions under which they migrate and include issues such as school bullying. Such experiences force these children to grow up amidst inequity and underestimation due to factors like poverty, cultural differences, and various expressions that may lead to stigmas, stereotypes, and xenophobia. These negative experiences can also result in the dismantling of the social fabric they had built in their country of origin,

making it difficult to rebuild these spaces in the new context. Such challenges can hinder the socialization processes necessary for their growth and the development of a healthy personality (Álvarez-Izazaga et al. 2022; Reyes Gutiérrez 2021; Estalayo et al. 2021; Romero-Rodríguez et al. 2021; Sultanic 2021; Brantl et al. 2021; Topalovic et al. 2021; Sanfelici et al. 2021).

Moreover, there are other noteworthy elements that arise in the context of adolescent migration, particularly in the case of Venezuelan adolescents migrating to Colombia and Latin America. These aspects include the loss of native culture due to fear of mockery by their peers, exposure to everyday situations that lead to early maturation to adulthood, and fundamental shortages such as "housing, food, clothing, medical care, hygiene services, telephone contact with relatives, counselling, and support to return to the localities of origin" (Ayala-Carrillo et al. 2013, p. 215; Zamora 2015).

This article explores the migration of unaccompanied or separated adolescents (UASA)[1] on the Colombian–Venezuelan border through the concept of a "social moratorium". This concept refers to the period of time granted by society to adolescents to postpone their economic, work, partner, pregnancy, child-rearing, and other responsibilities. This period allows them to enjoy a legitimate phase of study and training, facilitating physical and emotional maturation as they prepare for these future responsibilities (Villa Sepúlveda 2011). However, the violation of the rights of UASA and their premature entry into the workforce truncate this space for personal and social growth and development, impacting their prospective vision.

From this narrative perspective, it is important to contextualize the migratory phenomenon of adolescents in this border region, given that Venezuelan migration has been ongoing since 2016. This period marked the onset of massive waves of migration. These waves encompassed planned migrations, even among the highest social strata of the Venezuelan population, as well as migrations prompted by illness, violent events amidst protests, and increases in criminal activity or state repression.

The continuity of this migration can be attributed to political and economic factors. It was driven by the reduction in oil production and exports, hyperinflation, and frequent monetary reconversions, rendering survival and the ability to cope with extreme poverty unsustainable for the population. This economic strain was a major factor for the departure of the population of productive age. Consequently, migration patterns varied according to the age groups most affected and left without opportunities (Mazuera-Arias et al. 2019, 2020).

Notably, there was a predominance of first-order male migration followed by female migration (Albornoz-Arias et al. 2022). These migrations took various forms, with some individuals migrating alone, others in groups, and some as entire families. Since 2020, there has been a growing phenomenon of UASA. This has given rise to a new generation of migrants who navigate the desolate roads of many cities, either to leave Venezuela and reach Colombia as their destination or to transit through it in search of opportunities in other countries. The aggravating factor is that these migrants lack documents and economic resources for their journeys. They are unfamiliar with the geographic dimensions of the trajectories, the temperatures they will encounter, the risks of violence they will face, the cultural nuances, and the economic conditions of the host sites. Thus, they resort to illegal passage, as they have no adult guidance. This puts them at risk of being intercepted by the police and migration control authorities, who usually return them to their place of origin.

This migration imposes on the minds and life experience of adolescents a forced acceleration into premature maturity. They are compelled to assume economic responsibilities for the households they have left behind or for those where they seek to reunite with transnational relatives. However, during these reunions, they face the dangers of traveling to unknown territories, illness, the risk of dropping out of school, and exposure to the hazards that come with life on the streets. They lack resources, and their cognitive immaturity, characterized by traits such as unfocused attention, persistence of fantasy–reality thinking,

and a concrete thought process focused on the present (Piaget 1991), compounds these challenges.

The following section describes specific social contexts in which the migration of adolescents is occurring. It aims to provide insights into the unique realities they face in this migration process and emphasizes the urgency of addressing their well-being, which will also impact future generations.

Irregular migrant adolescents represent one of the most vulnerable human groups. In fact, they are three times more vulnerable: as adolescents, as migrants, and as persons in an irregular situation. These adolescents constitute a broad and diverse group that includes the following: (a) adolescents who arrive in the destination country to reunite with their families but do not participate in official family reunification programs or fail to obtain valid documentation through these programs; (b) adolescents who enter irregularly, either with one or more relatives or unaccompanied; (c) adolescents who have run away from their families and find themselves alone; (d) adolescents born in the receiving country but whose parents or guardians have an irregular status; (e) adolescents with regular status but living with parents or guardians in an irregular situation; (f) adolescents without residence permits or with expired visas; and (g) adolescents who are part of families whose asylum applications have been rejected (Ortega Velázquez 2015, p. 193).

These seven conditions encompass a detailed explanation of the daily challenges faced by irregular migrant adolescents. These challenges include the pain of uprooting, poverty, lack of familiarity with the new environment, separation from family members, and the premature burden of facing this reality. Notably, this population is on the rise worldwide. The United Nations Children's Fund (UNICEF) has reported that as of 2022, 37.9 million migrants are under the age of 20, comprising 14 percent of the world's migrant population. Among these, approximately 13 million include refugee children and adolescents, 936,000 are asylum seekers, and 17 million are children and adolescents forcibly displaced from their own countries (Fondo de las Naciones Unidas para la Infancia—UNICEF 2022).

In Latin America, the migration of minors reflects a dire situation where 6.3 million migrant adolescents in the region face life-threatening situations and multiple forms of violence. They are fleeing violent gangs that target them or escaping the poverty and exclusion that deny them opportunities and hope. Many travel north to reunite with their families, either alone or with acquaintances. They rely on dangerous routes and hire traffickers to help them cross borders. Deprived of their rights and needs, unprotected and unaccompanied, children and adolescents in transit become vulnerable to exploitation and abuse by traffickers and others (Fondo de las Naciones Unidas para la Infancia—UNICEF n.d.a).

The situation of Venezuelan children and adolescents in Colombia is not more encouraging. According to the Regional Inter-Agency Coordination Platform for Refugees and Migrants from Venezuela (R4V), as of October 2022, 2,477,588 migrants reside in Colombia (Plataforma de Coordinación Inter agencial para refugiados y migrantes de Venezuela—R4V 2022). They are exposed to forced recruitment, sexual violence, consumption of alcohol or psychoactive substances, living on the streets, sexual aggression, gender-based violence, and robbery or violent actions by criminal groups. These groups force and subject migrants to have sex with them or enter into child marriage or early unions (disappearance of children and adolescents), child labor in illegal mining, drug crops and laboratories, participation in illicit activities, participation in violent actions such as hired assassination, and drug trafficking. This has been highlighted in a study on the situation of refugee and migrant children and adolescents from Venezuela and their link to child labor in Latin America (Organización Internacional del Trabajo—OIT 2022).

The report on the characterization of migrant children and adolescents in Colombia[2] conducted by the Colombian Observatory of Migration from Venezuela has revealed that for 2020, more than 1.8 million Venezuelans resided in Colombia (Semana 2020). Of these, 36.4 percent corresponded to children and adolescents under the age of 18, totaling 998,000 minors (33 percent of children and 23.6 percent of adolescents), with "similar distribution

between men and women" (Instituto Colombiano de Bienestar Familiar—ICBF 2022, p. 19). Breaking down the age groups further, 42.5 percent of the minors were between the ages of 0 and 5 years, 33.4 percent were between 6 and 11 years, and 24.1 percent were in the adolescence stage (Instituto Colombiano de Bienestar Familiar—ICBF 2022, p. 26).

Studies have evidenced the situation of unaccompanied migrant children and adolescents in different Latin American countries, such as the triple vulnerability they experience in Peru: as minors; migrants and being involved in migration that leads to irregularity (Rivadeneyra Yriarte 2021); gender-based violence, human trafficking and smuggling, child labor, recruitment by armed groups in Colombia and Trinidad and Tobago (Durán Palacio and Millán Otero 2021; Palomo et al. 2022); documentation problems, overcrowding in institutional shelters and lack of resources to accompany minors in other stages of the integration process in Brazilian territory (Alves et al. 2019; De Moura 2021).

Of particular concern are the conditions of migrant children and adolescents without identification documents: 48 percent are aged between 0 and 5 years; 45.4 percent are aged between 6 and 11 years, and 38.1 percent are aged between 12 and 17 years. This lack of documentation has implications for school attendance, as 39.9 percent of the total number of children and adolescents do not attend any school. Those aged between 12 and 17 years are the most deprived of schooling, with a non-attendance rate of 50.1 percent. This is followed by children between 5 and 11 years of age, with a non-attendance rate of 34.2 percent. This educational deprivation prevents them from receiving school meals. In fact, 72.9 percent of those who are enrolled in school do not have access to school meals (Semana 2020).

Regarding access to healthcare, 76 percent of migrant minors are not affiliated with the healthcare system. This lack of affiliation varies across age groups, with the highest percentage of non-affiliation observed among adolescents aged between 12 and 17 years (83 percent), followed by children aged 0 to 5 years (76.9 percent) and children aged between 6 and 11 years (70.3 percent). This non-affiliation is due to lack of documentation (12 percent), lack of money (10.6 percent), excessive paperwork (7.3 percent), lack of awareness regarding the necessity of affiliation (1.7 percent), parents or caregivers not being affiliated with a company (0.6 percent), and limited interest or perceived need for healthcare services (0.3 percent). In addition, only 28 percent of migrant minors have access to vaccination and 38 percent to growth and development check-ups (Semana 2020). Adding to the complexity, children, adolescents, and the broader Venezuelan population have been grappling with food insecurity issues even before embarking on their migration journey to the Colombian–Venezuelan border region (Mazuera-Arias et al. 2021). Legally, the right to health is stipulated in Resolution 3384 of 2000 of the Colombian Ministry of Health, which aims to promote protective measures and prevent risk factors for preventable diseases among children and adolescents (Ministerio de Salud 2000).

Finally, a report by the Colombian Observatory of Migration from Venezuela has highlighted the situation of poor nutrition among children and adolescents. According to the report, 81.2 percent of migrant minors consume sugary drinks and 74.4 percent consume packaged food several times a day due to informality, scavenging, begging, or homelessness (Semana 2020). The combination of factors such as lack of access to healthcare, being outside of the education system, and the absence of a balanced diet creates a deeply concerning and challenging situation for migrant children.

The situation is made more discouraging by the fact that Colombia, as the receiving country, also grapples with issues of poverty. According to the Departamento Administrativo Nacional de Estadística—DANE (2021) the monetary poverty rate recorded was 39.3 percent, with 12.2 percent experiencing extreme monetary poverty. Multidimensional poverty, which considers indicators such as household educational conditions, children and youth conditions, health, work, housing conditions, and access to public utilities, was 12.9 percent, implying a structural poverty problem that affects nearly half of the country's population. The growing demographic increase resulting from Venezuelan migration exacerbates these issues. It increases unemployment conditions, the growth of illegal human settlements, informal work and, in general, social problems that have been accompanied

by xenophobia. All of these challenges are compounded by insufficient public policies and actions that fail to adequately address poverty among the native, returning, and migrant populations.

Therefore, it is the living conditions in the countries of the South that, in a certain way, confront the expectations of individuals and families seeking better opportunities. In many cases, the harsh reality of poverty in these regions is a direct result of social inequality, which not only widens disparities but also fosters misleading perceptions. These misconceptions serve as the fuel for these migrations, even though these individuals and families may face even more dire circumstances in the aftermath of the COVID-19 pandemic. The pandemic has had severe repercussions, including higher poverty rates and setbacks in economic, social, and human development, as highlighted by Comisión Económica para América Latina y el Caribe—CEPAL (2019). This shows that the number of poor people in Latin America rose to 209 million by the end of 2020, an increase of 22 million people compared with the value in the previous year. This situation underscores the existence of structural vulnerabilities, which is expressed in the intersectionality of the population already historically violated, as are the children and adolescents.

These structural poverties referred to above are leading migrant children and adolescents to desperately seek an occupation that allows them to contribute economically to their families. Some individuals and families facing challenging circumstances resort to desperate measures, such as attempting to build new homes or forming alternative family and social relationships. This can be seen as an attempt to fill the void left by the absence of stable family, social, and institutional support systems (Fernández García and Ponce de León Romero 2021). All of these are desperate decisions lead to the risk of losing their lives, early maturation, and loss of future opportunities as a result of the current lack of schooling.

The risk to the well-being and development of children and adolescents arises from the profound transformations occurring in their lives. These changes span various aspects, including the natural progression of physical and psychological maturity, alterations in their social interactions, and their ability to cope with the crises that emerge in the midst of the challenges they face. This perspective can be illuminated by referencing the work of Erikson (2004) on his theory of psychosocial development, which addresses the coping with crises typical of the ages. It is a gradual and progressive process that involves acquiring skills and resilience through the completion of age-appropriate tasks.

In the stages of trust–mistrust, autonomy–shame/doubt, initiative–guilt, and industry–inferiority that children and adolescents go through during childhood, they should surround themselves with quality caregivers who are present, participatory, and allow them to overcome crises and avoid long-term disorders (Arjona 2019). These caregivers help strengthen their work ethic through companionship, creativity, motivation, and common interests. This, in turn, helps them face new challenges, gain independence, and expand their capabilities and emotions, which are vital for their physical, social, and psychological maturation.

In addition, adolescents often grapple with role confusion as a result of social interaction through environments where there is a growing need to develop strategies for transitioning into adult life. During this phase, they explore their identity in terms of occupation, establish their values, and shape their personalities (Erikson 2004). This period serves as a crucial framework for adolescents to develop the skills necessary for forming their sexual, vocational, and value identities.

In this sense, Erikson (2004) coined the term "social moratorium" to describe the phase during which adolescents are tasked with discovering their motivations and making informed choices regarding their interests and passions in a mature manner, and scenarios such as school, sports training centers, and civic and ideological organizations are fundamental during this period. Zinnecker (2000) argued that this moratorium "Bildungsmoratorium" or "childhood moratorium" is a space of extension of the responsibilities of adult

life to build mature citizens, which are analyzed against various spheres of life (Côté and Levine 1988; Eichsteller 2009).

The omission of this social moratorium due to different situations such as war, extreme poverty, recruitment into armed groups, sexual exploitation, and premature maturation to assume adult roles causes identity problems, social and personal anomie, loss of trust in institutions, and change or loss of work vocation (Vera Noriega et al. 2013). Therefore, it is warned that "adolescence is a time of opportunities or risks. Adolescents are on the verge of love, a life of work, and participation in adult society" (Papalia et al. 2019, p. 514). During this period, they can either channel their identity and occupational aspirations or face conflicts that may result in the development of addictions, early pregnancies, or antisocial and criminal behavior (Erikson 2004).

The situation of adolescents facing the responsibilities of adulthood without being fully prepared presents a significant challenge, particularly for UASA who are forced to take on new roles for which they are not prepared. This interrupts their social moratorium period, resulting in psychoemotional and social changes that affect the free development of their identity. This identity, far from having a promising future for them, offers them the immediate satisfaction of some basic needs, sacrificing their integral psychosocial development. This results in issues including identity confusion, impulsive behavior, a lack of matured aptitudes and skills, emotional instability, and challenges in forming secure attachments and cohesive group relationships.

The situation of children and adolescents in the condition of unaccompanied migrants means they are susceptible to losing their childhood and adolescence stage to study, mature and assume responsibilities and roles that correspond to adulthood, and it is difficult for them to define their personal identity due to their incorporation into informal labor activities. These realities are evidenced in several studies among them in Italian, mestizo and migrant families (Crocetti et al. 2011); urban youth in Ghana (Langevang and Gough 2009); ethnic and racial identity during adolescence and early adulthood in the United States (Umaña-Taylor et al. 2014), among others. It should be noted that no studies were found on the social moratorium in unaccompanied Venezuelan migrant children and adolescents, which is why this article represents a contribution.

These structural vulnerabilities framed in social dynamics make visible the transformation of processes, lives, and institutions such as the family (Aurioles-Tapia and Torres-López 2014). These vulnerabilities also indicate the negative changes typical of a liquid (Bauman 2015) and empty society (Lipovetsky 2006) framed by individualism; the irrational use of technology; hypersexuality; and various dangers such as sexual violence, human trafficking, sexual exploitation, domestic slavery, child marriage, pregnancy, mental health issues, loss of life's meaning, suicide, and substance abuse. Because of their status as unaccompanied or separated migrants, they often assimilate into a situation where they become disconnected from essential elements like protection, rights, access to education and healthcare, and their long-term prospects for development in their host country.

Accordingly, this article aims to expose the Venezuelan migration of UASA at the Colombian –Venezuelan border through the concept of a social moratorium.

## 2. Materials and Methods

We conducted our research using a qualitative approach with an inductive purpose, aiming to contextualize the reality of UASA from Venezuela who have arrived in the city of Cúcuta, Norte de Santander, Colombia. Our goal was to understand the experiences that add complexity to their reality, including their interactions, decision-making processes, access to essential needs, and expectations (Silva Batatina 2017; Delgado 2010). Consistent with this intentionality, we used the narrative method, allowing us to delve into the experiences of these adolescents as they navigate a sociohistorical and psychosocial way of life. Through brief accounts of their lived experiences, they sequentially recounted episodes of their subsistence during migration, their arrival, and their stay in the country. These narratives of everyday life offered valuable insights into their experiences, including their

sense of time, familiar places, and the significance they attributed to their lived experience (Muñoz-Proto et al. 2020).

As part of the narrative methodological process we employed to consecutively order the narratives of the adolescents throughout their migratory process and current settlement, we created a sequencing matrix. This matrix was designed to organize the category of social moratorium based on the theoretical analysis of Erikson (2004) within the context of unaccompanied or separated migration. Following the guidance of Sparkes and Devís (2007), our objective was to formulate guiding questions that would lead to a deeper understanding of the "what" and "how" of migration and the new context to identify "common themes in the stories or narratives with the intention of arriving at certain generalizations", where our goal was to ask guiding questions that would lead us to understand the experiences and dangers faced, to identify "common themes in the stories or accounts with the intention of arriving at certain generalizations" (p. 53).

As the realities that did not allow this postponement of work commitments, questions were asked such as: "Tell us how did the idea of migrating come up, how did you make that decision, with whom did you decide to travel, and how was the journey of this migration until arriving in Cúcuta, do you plan to continue the trip, how far was the process of arriving in the city of Cúcuta, and what experiences have you had to live here, tell us how do you face the different situations that you live here, how do you face the different situations you live here?" We selected places of concentration of migrant populations in the city of Cúcuta, such as the Mercado de la Sexta, Mercado de Cenabastos, and Avenues 7 and 8 near the Transportation Terminal of the border city of Cúcuta, Norte de Santander, Colombia.

Given that the participants were adolescents (Table 1), we selected the group interview as the data collection technique. The technique allows them to discuss their experiences in an open, free, and natural way and with collective experiences. As proposed by Amezcua (2003), "It is synonymous with group conversation, informal, and in situ. It refers to that which arises spontaneously when the researcher goes in search of informants in the field and finds them grouped in their environment, engaging them in an informal conversation" (p. 113). During the study's development, we formed affinity, considering that they work in the same street, they recognize each other and cooperate in their daily dynamics, forming groups of 7 members in each of the markets and of 5 members in Avenues 7 and 8. This affinity avoided establishing the further selection criteria that they are minors and that they have migrated alone, so they are not grouped by gender, age, life experiences, although this was found as the group interviews are carried out.

At the sites, we read the informed consent form to the unaccompanied adolescents, and all agreed to participate in the study. We guaranteed the confidentiality and anonymity of the informants. No names or identifications were requested, and there were no risks concerning the participation of the adolescents in the study. We carried out the primary data collection in November 2022. We did not have the authorization of their parents or legal representatives because these immigrants undertook the migration process alone, i.e., the reins of their lives were not in the hands of parents or caregivers at that time. However, the participation of adolescents in this study contributed to making this reality visible to academia and decision makers in the host country. Next, we proceeded to collect narratives in the group interviews with the adolescents, engaging in active listening throughout the process. Additionally, we maintained narrative field diaries to capture the experiential expressions that encompassed their thoughts, emotions, and feelings. These expressions were instrumental in understanding how they were experiencing and constructing their personal histories. These diaries served as a means to prefigure or experience the configuration of narratives that would later become a part of their biographical history, contributing to their overall self-identity and self-configuration (Delory-Momberger 2015). The primary information collected is available in an open-access repository (Ramirez Martinez et al. 2023).

**Table 1.** Sociodemographic characteristics of key informants.

| Code | Sex | Age | Nationality | Place of Origin | Place of Final Destination |
|---|---|---|---|---|---|
| Interviewed at the Mercado de la Sexta | | | | | |
| A01 | F | 13 | Venezuelan | Caracas, Capital District | Migrating to Bogotá |
| A02 | F | 14 | Venezuelan | Portuguesa State | Migrating to Bogotá |
| A03 | M | 17 | Venezuelan | Portuguesa State | Migrating to Bogotá |
| A04 | F | 15 | Venezuelan | Nueva Esparta | Cúcuta |
| A05 | F | 17 | Venezuelan | Aragua, Maracay | Cúcuta |
| A06 | F | 17 | Venezuelan | Carabobo, Valencia | Cúcuta |
| A07 | F | 17 | Venezuelan | Carabobo, Valencia | Cúcuta |
| Interviewed at the Mercado de Cenabastos | | | | | |
| A08 | M | 14 | Venezuelan | Apure State, San Fernando | Cúcuta |
| A09 | M | 14 | Venezuelan | Trujillo State | Cúcuta |
| A10 | M | 15 | Venezuelan | Trujillo State, Sabana de Mendoza | Cúcuta |
| A11 | M | 17 | Venezuelan | Trujillo State | Cúcuta |
| A12 | M | 15 | Venezuelan | Trujillo State, Valera | Migrating to Barranquilla |
| A13 | M | 17 | Venezuelan | Trujillo State, Valera | Migrating to Barranquilla |
| A14 | M | 17 | Venezuelan | Trujillo State, Valera | Migrating to Tibú |
| Interviewed on Av. 7 near the transport terminal | | | | | |
| A15 | M | 16 | Venezuelan | Lara State | Cúcuta |
| A16 | M | 16 | Venezuelan | Carabobo State, Valencia | Migrating to Perú |
| A17 | M | 16 | Venezuelan | Cojedes State, San Carlos | Migrating to Bucaramanga |
| A18 | M | 17 | Venezuelan | Anzoátegui State | Cúcuta |
| A19 | M | 15 | Venezuelan | Carabobo State | Cúcuta |
| Interviewed on Av. 8 near the transport terminal | | | | | |
| A20 | F | 15 | Venezuelan | Carabobo State, Guacara | Cúcuta |
| A21 | M | 16 | Venezuelan | Lara State | Cúcuta |
| A22 | M | 17 | Venezuelan | Portuguesa State | Cúcuta |
| A23 | M | 17 | Venezuelan | Cojedes State, San Carlos | Cúcuta |
| A24 | M | 17 | Venezuelan | Caracas, Capital District | Migrating to Bogotá |

Source: Prepared by the authors.

To analyze the narratives, we used the "paradigmatic content analysis" proposed by Sparkes and Devís (2007). In this approach, "similarities and differences among the narratives are examined in order to develop general knowledge about central themes that constitute the content of the stories under study" (p. 50). The division of the text into units was proposed for the hermeneutic analysis. This was confirmed by Suárez Relinque and Moral Arroyo (2020). The authors guided the development of segmentations, codifications, families, conceptual networks, and information maps, which were developed through the use of the Atlas.ti 9.0 software. This approach helped identify three elements influencing the early abandonment of the social moratorium: 1. the violation of rights, 2. the working life, and 3. the confrontation of dangers.

### 3. Results and Discussion

Analyzing the narratives of the 24 unaccompanied migrant adolescents who arrived in the city of Cúcuta from Venezuela, 8 migrated alone, making friends during the trip to keep them company. The remaining 16 traveled with their siblings or cousins, who were also minors. Of these 24 migrant adolescents, 7 were females and their ages were as follows: 13 years (1), 14 years (1), 15 years (2), and 17 years (3). Of these 24 migrants, 17 were males aged 14 years (2), 15 years (4), 16 years (4), and 17 years (7).

Regarding the link to the school system, they were not interested in accessing the educational system, as they were seeking employment and economic opportunities. Among the 24 adolescents in our study, 2 completed up to 6th grade, 11 completed 1 year of high school, 5 completed up to 3 years of high school, 1 completed up to 4 years of high school, 2 completed up to the first year of high school, and 3 were high school graduates in validation mode. None reported having documentation accrediting their studies. All of them stated that they had been out of school for approximately three years. Some felt unsafe in Venezuela since they were recruited by criminal gangs, and others stated that there was no electricity in their schools. Most of them had dropped out of school due to the multiple shortages of their basic needs, resulting in seeking work to pay for them and their families.

The results presented below detail the elements that impede the social moratorium of UASA in Venezuela. These are presented by explaining: 1. the violation of the rights of adolescents, 2. working life, and 3. the various dangers they face.

### 3.1. Violation of the Rights of Unaccompanied and Separated Migrant Adolescents (UASA)

In response to the first question that sought to identify the reason for migration, the 24 adolescents agreed that they immigrated because of extreme poverty, hunger, and few opportunities for their families, and all 24 agreed that some of their relatives had already migrated, so they follow the same trend, in addition to stating that in Venezuela, they have been left without many of their relatives and friends. In the coding of this emerging category, there are five elements to be highlighted, as shown in Table 2:

**Table 2.** Emerging category and subcategories of the violation of rights.

| Category Code: VR | Codes for Emerging Subcategories |
|:---:|:---:|
| | DV 1 No access to food |
| | DV 2 No access to housing |
| Violation of rights | DV 3 School dropout |
| | DV 4 Health risk |
| | DV 5 Lack of leisure time |

Source: Prepared by the authors.

The reality of migrant adolescents reveals a growing social problem after the repeated violation of their rights. In its report, UNICEF stated that the "History of children's rights" highlights advances through conventions that have allowed for recognizing the role of children as social agents and promoting legislation to prevent juvenile delinquency, child labor, and forced recruitment in armed conflicts, as well as to put an end to the sale, exploitation, and sexual abuse of children and adolescents (Fondo de las Naciones Unidas para la Infancia—UNICEF n.d.b).

However, the reality of the adolescents shows that the Colombian–Venezuelan border is far from this protection. It was found that 11 adolescents were living on the streets, 4 in temporary shelters, and 9 in the homes of Venezuelan people they met upon arrival or with a distant relative they identified as a support network. However, in all the conditions described, they reported contexts of overcrowding, hunger, lack of cleaning supplies, poor food, and excessive work in terms of their age. Some individuals faced the difficult situation

of not having continuous housing, as they were required to pay for it on a daily basis. When they lacked the necessary economic resources, they found themselves without shelter and had to sleep on the streets (Table 3).

**Table 3.** Lack of access to housing and protection.

| | |
|---|---|
| A20 | "It was a very difficult week walking, sleeping on the streets, spending time under the bridges, and we arrived here and it is the same, spending time recycling." |
| A21 | "I've been on these streets for a few days now, here I recycle, I ask for help, I get by with the locals around here, they know that we're migrants, and that we're looking for opportunities." |
| A22 | "We have been on the streets in these parts, it has been hard, days of hunger and tiredness." |
| A23 | "Here sleeping on the streets, looking for food, recycling and washing dishes in a restaurant to get food. Now I went to an international organization and they are going to help me with the tickets to return to Venezuela." |
| A24 | "Now I'm here on the street these days, while they are looking for a truck to take cargo to Bogotá and leave at dawn, to let us travel as stowaways." |

Source: Prepared by the authors.

When asked what they most needed, they stated cleaning supplies, connectivity with their family in Venezuela, food, support to get a job, and resources to continue their journey (four expressed their desire to travel to Bogotá, the capital city of Colombia, because they believed they would be able to find work there easily). Furthermore, two wished to travel to the coast of Colombia, to Barranquilla, stating that they had heard it was easier to find work there and that it provided a better pay. One wished to go to Bucaramanga, another wanted to return to Venezuela, and another wanted to go to Tibú, a municipality in Norte de Santander, because "he heard that he can be employed on a farm there" (A14).

Two female adolescents were in a situation of commercial sexual exploitation of children[3] because in their street condition. They had made networks with other women in their country. They found prostitution to be the only way out of subsistence; thus, they had rented a room where they lived and made payments jointly to reduce costs. They found themselves in a situation where securing the bare essentials for survival kept them from pursuing other opportunities for assistance and more desirable futures. This choice was often influenced by the deep emotional bonds and shared identity they had with the individuals they lived alongside.

The early initiation of adult life was also manifested by an adolescent girl who lived with her boyfriend and did not know or have access to family planning methods. She did not understand or visualize her future reality but naturalized this form of subsistence in the creation of a family life as a form of support.

Finally, in this description of the current lifestyles of the adolescents, it should be noted that, although they undertook trips to reunite or to obtain resources to send to their families in Venezuela, the context had confronted them with intra-family violence or mistreatment with the relatives they met, leading to painful separations and greater psychological affectation and loss of protective environments, as reported by the adolescents (Table 4).

All of these life situations were related to the hard experience, emotional overload, family breakups, and the psychological damage to which adolescents were exposed, as a consequence of "deficient socioeconomic situation in relation to family stressors and social support, serious situations of socioeconomic vulnerability, feminization of care, polyvictimization of children, high number of stressors, and low perception of social support" (Fernández Rodríguez and Cracco Cattani 2022, p. 97). This situation impacted emotional disorders in children and adolescents due to separations from home, new roles, and situations of fear and uncertainty (Bragado Álvarez and García-Vera 1998), leading to

the anguish of "getting resources to send my mom" (A10) and "looking for a job to be able to bring food to my parents because in Venezuela there is nothing" (A12), as well as the food needs they face as described in the Table 5:

**Table 4.** Violence in migrant families.

| A03 | "… my relationship with my older siblings, who are in Bogotá, is not very good…" |
|---|---|
| A02 | "… my relationship with my mother is one of conflict and she has no knowledge of the trip…" |
| A09 | "… about three weeks ago I stopped living with my mother because she didn't work and even though I brought her what I could get, we had a bad relationship and I was mistreated… she hit me a lot and treated me very badly, that's why I left her side. A few days ago I went to look for her and she no longer lived in the same house, she kept my clothes and my papers." |
| A23 | "… together with my uncle we walked along the route from Los Patios to Pamplona and from there to Bucaramanga, then to Manizales, there I worked for three months picking coffee and in a hardware store, however, I received bad treatment from that uncle and I decided to return to Cúcuta, in a truck[4], as stowaway[5], they stole all my belongings and now I only have the clothes I have on." |

Source: Prepared by the authors.

**Table 5.** No access to proper food.

| A01 | "You can endure a lot of hunger and you have to look for it." |
|---|---|
| A01 | "We go from office to office asking for help, they give us food." |
| A03 | "In the field, in the crops, but I only had food and sleep." |
| A05 | "Everyone is very poor in Venezuela, there is no food." |
| A08 | "Your meals? Whatever they give me, I ask for food or money." |
| A09 | "With that I paid the rent and food, but my mother didn't work." |
| A09 | "But if they ask me for food, I give from here what I collect, rice, lentils." |
| A10 | "There is a lot of hunger." |
| A12 | "One gets tired of not having clothes, there is no food, there is nothing there." |
| A12 | "But there are days when they are not done because the food is done, so we have to sleep on the street." |
| A13 | "It was all day and they gave us three meals of pure soup." |
| A14 | "Food can be once a day." |
| A15 | "There's no food." |
| A15 | "To see what they give me, mostly food for the day so I don't miss out." |
| A16 | "Like to look for food." |
| A18 | "One meal a day." |
| A18 | "We share what they give us for food." |
| A20 | "There are times when you can't make food." |
| A22 | "But there we only do it for food." |
| A23 | "Looking for food, recycling." |

Source: Prepared by the authors.

In addition to the housing and food shortages described above, there was also a lack of access to leisure time that allowed them to isolate themselves from these stressors and achieve moments of well-being (Table 6).

**Table 6.** Leisure time.

| A | VR5 Lack of leisure time |
|---|---|
| A01 | "Most of the time we have been on the streets." |
| A02 | "We have had nowhere to stay, here on the streets, you walk and see what you do." |
| A06 | "You never let go of your suitcase, it's ready to go because if you don't do what you do at night, you can't sleep but on the street." |
| A07 | "So as not to be on the street and to be able to eat and live under a roof. We are already saving to bring my mother and then we can bring my sisters." |
| A08 | "That's where we came to live on the street, looking for something to do, begging, selling some candy." |
| A08 | "Since we left, it has been hard, walking, sleeping in the streets, wherever we could find, and we passed the trails to get to Cúcuta to continue living in the streets." |
| A08 | "There I go out from 10 o'clock in the morning to sell these sweets and I return late at night with what I sell." |
| A10 | "Sometimes I work in a hardware store, cleaning plastic refrigerators, and now I'm still looking for work, because here I sell candy, but I don't earn anything, I want to work in whatever I can get." |
| A11 | "We rent a room for 5000 pesos a night, but there are days when we don't make it because we have to make the food, so we have to sleep on the street." |
| A13 | "It is also hard, there is no way to sell something good, there is no work, and staying on the street is also bad." |
| A14 | "Well, we rent a room and there we have a small stove, for which they charge us 15,000 pesos a day, with my sister we make cakes and then we go out to the streets, my sister with her cakes and I sing at the traffic lights with my brother-in-law where we are from 7 am to 7 pm." |
| A15 | "We are all in the same situation, on the streets there are a lot of fellow countrymen and women—you know what I mean—we have to ask them and move around." |
| A18 | "Here on the street, we don't know anyone, we go over here, help the other kids and see who helps us and accommodates us." |
| A18 | "Here on the street, we wait for the sun to rise to ask for help from our friends, some stay here too, there are days, some sell and do what they need to do, others go to their rooms, others stay on the street, we look for where it is safer." |
| A20 | "My father used drugs and we were in need every day, that's why my father decided that we should come and we came here to the street, but the fights are getting harder and harder and now I'm not with them, I'm with a lady and her daughters who are here doing sex work and they help me and let me stay with them." |

Source: Prepared by the authors.

This lack of dignified treatment, among other issues, is referenced in the conventions for the protection of children's rights, such as the United Nations Convention on the Rights of the Child, the International Convention on the Elimination of All Forms of Racial Discrimination, the Hague Convention on Protection of Children and Cooperation in Respect of Intercountry Adoption, the International Convention on the Protection of the

Rights of All Migrant Workers and Members of Their Families, Political Constitution of Colombia, Code of Childhood and Adolescence, Law 1098 of 2006, and Law 679 of 2001. This law aims to prevent and counteract exploitation, pornography, and sexual tourism with minors. It establishes sanctions for those who commit these crimes. They show the violation of the rights of UASA.

In addition, Law 1336 of 2009 intends to prevent and punish acts of discrimination against adolescents, while Law 1616 of 2013 establishes the National System for School Coexistence and Training for the Exercise of Human Rights, Education for Sexuality, and the Prevention and Mitigation of School Violence. It aims to promote peaceful and respectful coexistence in educational environments. Law 1804 of 2016 establishes measures to prevent sexual abuse against minors and increases penalties for those who commit these crimes.

### 3.2. Working Life of Unaccompanied and Separated Migrant Adolescents

In the Colombian legal framework, the Substantive Labor Code (Congreso de Colombia 1951), in various articles such as Article 161, states that minors between 15 and 16 years of age may work up to 6 h a day, 36 h a week, until 6 pm; minors under 17 years of age may work up to 8 h a day and up to 48 h a week, until 8 pm; and children between 5 and 14 years of age may work if justified by some cultural or sports activity but not exceeding 14 h a week. Likewise, the activities may not involve danger, be harmful to health, or threaten physical integrity.

In this same Substantive Labor Code, the prohibitions are made clear, as in Article 171 where the following is prohibited based on age: minors under 14 years of age may not work in industrial companies or in agricultural companies when their work prevents them from attending school; minors under 18 are prohibited from working at night, in conditions hazardous to health, or as stokers. Articles such as Article 30 make clear the need to have the written authorization of the labor inspector or, in their absence, of the first local authority, at the request of the parents and, in their absence, of the family ombudsman.

Article 59 makes it clear to employers of adolescents under 18 years of age the prohibition on transferring them from their place of residence to perform any act that violates or threatens the physical, moral, or psychological health of the working minor. Article 242 prohibits dangerous, unhealthy, or strenuous work.

These prohibitions and labor frameworks are ratified in the Code of Childhood and Adolescence, given that Article 35 ratifies compliance with the Substantive Labor Code (Congreso de Colombia 2006).

These are not considered by the adolescents who describe realities that ignore the legal context that provides them with rights and are not applied in the contexts of the city of Cúcuta, its streets, and the places where the work developed by the adolescents interviewed is described.

In these 24 narratives of life experiences, it is important to highlight the motivation of the trip in a labor sense, as it is the only possibility of subsistence and sending remittances. Hence, the most emphatic aspect of their actions is their pursuit of employment, which becomes a pivotal factor influencing their choice of destination, the relationships they establish, and their social networks. This relentless quest often leads them to undertake perilous journeys, traveling without a legal representative and resorting to illegal passages. These actions expose them to various forms of violence, including labor exploitation, begging, and recruitment into criminal activities. These dire circumstances arise from their inability to access basic necessities such as food, shelter, personal hygiene, education, and legitimate employment opportunities.

Here, subcategories emerged as follows (Table 7):

**Table 7.** Emerging working life category and subcategories.

| Category Code: WL | Codes for Emerging Subcategories |
|---|---|
| Working Life | WL1 Poverty |
| | WL2 Abuse |
| | WL3 Exploitation due to poor remuneration and excessive workload |
| | WL4 Non-definition of functions and schedules |

Source: Prepared by the authors.

The predominant and structural reality of the Venezuelan migration problem is poverty, and the adolescents recognize it in their personal and family lives, which is the trigger for migratory journeys as well as the dangers assumed, as revealed in their narratives in the Table 8.

**Table 8.** Poverty experienced by adolescents.

| A | WL1 Poverty |
|---|---|
| A02 | "Things in Venezuela are not good, there is hunger and poverty." |
| A05 | "Well, my parents' situation is not good, in Venezuela there is a lot of poverty, there are no consistent public services." |
| A06 | "Well, in my house everything was very hard, there is nothing to do to earn money, there is a lot of poverty and needs of everyone in the house, in the families, in the neighbors, we all have a hard time and one sees." |
| A07 | "There is hunger and poverty." |
| A13 | "There is a lot of poverty and need in Venezuela, there is nothing. There I lived alone because my father died about three years ago, my mother found a new partner." |
| A14 | "Well, with the situation there, you know, well, poverty, no electricity, no gas for a long time, water is very rationed, food can only be once a day." |
| A15 | "There is no money, there is no food, there is no electricity, there is no food and no gas to cook with, there is a lot of poverty." |
| A18 | "There is a lot of poverty and that is when we started with a cousin to say that we had to come here, that we have to look for a way out and here there are more opportunities." |
| A22 | "The situation is very bad. Seeing that there is nothing to do and there is nothing but poverty." |

Source: Prepared by the authors.

Poverty often blinds adolescents to their current circumstances, driven by the goals they have set for themselves. They aspire to work and send resources back to their families, aiming to reunify with their loved ones and provide essential economic support. Consequently, some of them believe that leading a clandestine life is the only way to avoid being returned to Venezuela by the police and Colombian migration agents. This is why they opt to live on the streets rather than seek refuge in shelters. In this regard, Glockner Fagetti and Álvarez Velasco (2021) argued that the current conditions of migration make it necessary to stop seeing children and adolescents without their reflective, interpretative, and self-perceptive capacity to participate in social life. Such complex processes of migration should call for the creation of a culture of childhood policy, where their processes, values, and participation are recognized to free them from the structural poverty they face, from the situations that the system itself does not avoid and leaves them to face, after pursuing a working future that vanishes do to not having policies that include them, and stop looking at them in the light of parents or caregivers who are not and will not be by their side due to migratory circumstances.

Another element that emerges is the mistreatment suffered by adolescents in their families, which confronts them with the reality of stressful contexts and lifestyles that make them transform their bonds and caregiving responsibilities, as highlighted in the Table 9.

**Table 9.** Forms of abuse in adolescents.

| A | WL2 Abuse |
| --- | --- |
| A09 | "I stopped living with her because she beat me a lot and treated me very badly." |
| A15 | "My dad used drugs and we were in need every day, so my dad decided that we should come and we came here to the street, but the fights are getting harder and harder and now I am not with them." |
| A11 | "I go there to recycle and the guys harass me, one day a man saw me recycling and I was using a knife to cut the cardboard and cut the recycling, and he took the knife away from me and punched me 10 times." |
| A24 | "Now here on the streets these days, while they are looking for a truck to take cargo to Bogotá and leave at dawn, to let us travel as stowaways." |

Source: Prepared by the authors.

Situations that result in the expulsion of adolescents, leading them to assume working conditions, even if they are not dignified and regulated in the country are shown in the Table 10.

**Table 10.** Exploitation due to poor remuneration.

| A | WL3 Exploitation (Poor remuneration) |
| --- | --- |
| A01 | "Here, to sell, to beg, to visit international organizations." |
| A03 | "We can sell candy, vegetables, or small things or work in restaurants, washing dishes or waitressing." |
| A04 | "We work with men, you know what I mean." |
| A07 | "Well, with my boyfriend we know how to make shoes and that's where we go, the pandemic doesn't leave much, but there are those who wait for the season and little by little we do it and we complete it here in the market, by the day, in restaurants and when I don't go out, I help sell fruit." |
| A13 | "Selling sweets." |
| A14 | "I've been working on a farm belonging to one of his brothers-in-law." |
| A15 | "I recycle, I ask for help, I offer the candy, but sometimes they don't buy, they give the money." |
| A18 | "Selling water." |
| A22 | "I am a recycler, starting in the morning I look for plastics, cartons, cans, bottles, here in the garbage or thrown from the buses, those that leave the terminal, in the cafeterias, there are possibilities, what happens is that there are many in the same thing." |
| | "Helping in mechanics around here, I sell water." |
| A22 | "Now I have just arrived from Caldas, there I was working in the market, in the crops, picking coffee, whatever I can get, but they don't pay well, so I came back to look for work here." |
| A23 | "Oh no, that was a lot of walking and joining small groups that we knew from other kids, getting rides in carts." |

Source: Prepared by the authors.

Unaccompanied Venezuelan adolescent migrants, despite their inexperience and possible maturity to assume responsibilities, are aware that a difficult working life awaits

them, as the findings in Ecuador (Herrera Mosquera and Pérez Martínez 2021), Brazil (Riggirozzi et al. 2023), and Colombia (Nagle and Zarama 2021) show.

The informal labor conditions starkly illustrate that the regulated framework for employment is far from the reality experienced by adolescents. Ochaíta et al. (1999) argued that avoiding dangerous jobs and promoting beneficial jobs in which the length of time, schedules, activities performed, age of the adolescents, remuneration, responsibilities, etc., are controlled determine whether the results will be stimulating for physical, cognitive, and social development or, on the contrary, undermine their dignity and self-esteem, as can be seen in the stories of the Venezuelan adolescents analyzed.

Another reality that influences the labor condition of child labor exploitation in adolescents is the lack of working conditions, as referenced in the Table 11.

**Table 11.** Lack of definition of labor functions for working migrant adolescents.

| A | WL4 No Definition of Duties and Hours |
|---|---|
| A09 | "From 10 o'clock in the morning I sell these candies and I come back late at night with whatever I sell." |
| A13 | "We asked for work and they told us they were going to pay us 60,000 pesos in a week, but in the end they only paid us 10,000 pesos, but the work was very hard, we got up early from 3 in the morning until 10 at night and all day long we had to sweep, set up the tables, clean the kitchen, and provide service. That's where we came to Cúcuta." |
| A17 | "Here in the mornings I help to sell whatever comes out in the restaurants or bakeries, sell water, clean refrigerators, throwing out trash, like looking for food and wait for it." |
| A21 | "I am a recycler, starting in the morning." |

Source: Prepared by the authors.

The conditions described above by the migrant adolescents in this study show that this situation dates back to the beginning of the 20th century, when countries did not have child protection standards, causing children to work in parallel with adults in unsafe conditions, especially for children from poor families where childhood ended up becoming shorter and shorter or non-existent, as they worked inside and outside the home (De la Fuente Núñez 2021).

Working together between governments, society, and international organizations for the eradication of child and adolescents labor should be in everyone's interest, given that due to its characteristics it represents overexploitation, abuse, humiliation, dehumanization, and risk, which question us socially due to the biological and psychosocial consequences, especially for children (Leyva Piña and Pichardo Palacios 2016). This eradication implies addressing informal and agricultural jobs where long working hours are normalized, as well as the expansion of legislation and new concepts that adjust to the social reality of migration and its transnational implication to address the complex demands of the situations experienced by migrant populations, depending on the combination of their relationship with the society of destination and origin, frequency of movement, living conditions, administrative situation, perception of the population, and interaction patterns (González-Rábago et al. 2021, p. 82). This becomes urgent and unites us in terms of this call and awareness.

### 3.3. Dangers Faced by Adolescents

This segment will describe a condition that highlights the situation of cognitive immaturity of migrant adolescents, which is gradually confronted with the reality of the journey and the conditions of hunger and work that lead them to face this harsh reality. These accounts of their experiences highlight hopeful looks and tones of voice full of energy and dreams, the idealized—but fair—goals that should be a reality in their lives, which are intertwined with a painful ending to their narratives, which end up showing this transition

to an adult life that comes with toil and pain, as evidenced by the narratives included in Table 12.

**Table 12.** Emerging category and subcategories of working life.

| Category Code: CD | Codes for Emerging Subcategories |
|---|---|
| | CD1 Sexual exploitation |
| Coping with Dangers | CD2 Robbery |
| | CD3 Consumption of psychoactive substances |

Source: Prepared by the authors.

They referred to the experience of multiple unpaid jobs, robberies, and other situations as if they were an adventure that they make magic after overcoming it. For example, listening to three of them, who traveled from Venezuela and who say that their "purpose in Colombia is to come to work with their brother and cousin to raise money and buy dollars to set up a record label and start recording a demo" (A17), "to find a job and be able to buy clothes, food, and meet all the needs of my family" (A04), and "to work to bring my mother and travel together to another country" (A05). Equally hopeless for an adult, but not for them, is to hear their stories where they tell of the journey, the hours of intense walking in which they do not give up and show their strength and courage: "We crossed the trail and I had to cut my hair to sell it to pay for the border crossing" (A05), "on the trail they stole my bag where I had my documents" (A06), and "about a month ago I left Pasto where I was working and they never paid me" (A03). These narratives reveal expectations that intersect with the realities of living on the streets, scavenging, illegal groups in the expectation of their confinement, for which the development of comprehensive protection policies that guarantee them food, housing, education, "emotional care and protection against any form of violence or abuse, guaranteeing the exercise of their rights" are urgently needed (Instituto de Políticas Públicas en Derechos Humanos del MERCOSUR 2019, p. 39).

These contexts, although they place the adolescents in a place of victims, facing migratory challenges, new cultures, separation, and need, are among the many changing realities that affect their mental health, as they continue their life process, facing subsistence and trying to adapt, learning before and during the migratory transit, as their stories continue:

> "to get to San Antonio[6] we paid a guard 50 thousand sovereigns[7], on the way we got off at the alcabala of El Vigia[8] and the driver helped us to pass because they were going to return us for being minors, there we arrived in San Antonio, we slept in the terminal where we met the trocheros[9] who offered to help us cross the border in exchange for carrying some bags, on the way the paramilitaries[10] checked what was in the bags and there we realized that it was coffee, finally we arrived at La Parada and we only had 70 thousand sovereigns which at the exchange rate was 2600[11] Colombian pesos. We finally arrived at La Parada and only had 70 thousand sovereigns, which at the exchange rate was 2600 Colombian pesos. We collected 1000 more pesos that a lady gave us, and with that we went by bus[12] to the terminal. When we got there, a pirate[13] cab gave us a free trip[14] to the curve[15], and there we worked in a restaurant where we were initially told that we would be paid 60[16] thousand pesos in a week, however, in the end they only paid us 10[17] thousand pesos, the work schedule was from 3 in the morning until 10 at night. We had to sweep, set up the tables, clean the pigsty[18], and serve. After that, we went back to Cúcuta and decided to start selling water in the street. We paid 5000[19] thousand pesos a day for lodging, but sometimes we could not get the money together and we slept on the street. Sometimes in June or July when we collected some money for the family we go to Venezuela to bring

them something, but we go back to Colombia to work because the situation in Venezuela is difficult". (A01)

Another adolescent said that it has been difficult to stay on the street because he has had to see how sexual services are requested and he witnessed constant drug use, in addition to the high temperatures of 35 °C to 37 °C, hunger, and other situations that the street presents in this scenario of the capital city of the department where there are various crimes and illegal forms of smuggling, theft, trafficking, and other social problems that have characterized the city (A24). Faced with these dangers and risks, morality and social interaction stand out, highlighting the importance of empathy, perception of others, and especially the questioning of moral dilemmas, with important arguments for the formation of prosocial moral reasoning, since this provides the ability to mature in the face of events in which needs and desires are confronted after prohibitions and minimal punishments. Therefore, it is worth thinking about these perspectives of migrant generations who are abused, not defended, exploited at work and in street conditions and family conflict, struggling to grow in love and family work, but feeling violence in their homes and environments, which makes the possibility of social care for migrant adolescents even more urgent, as a sense of urgent humanization (Retuerto Pastor et al. 2004).

This is a context of the Colombian country, in which El Espectador (2021) records that "the national coverage of Venezuelan migrant minors of school age enrolled in Colombia was 203,494. In contrast, 80.506 (28.35%) migrant children and adolescents were un-enrolled and without access to a quota", which evidences lack of opportunities, a migratory dynamic that is based on constant mobility, lack of routes and dialogues between governments of Venezuela and Colombia and between other countries that allow the location of relatives of minors, in order to promote family reunification, humanitarian transport, "follow-up after family reunification, in the face of difficulties in family reunification, adoption that allows them to have Colombian nationality (as the case may be), and that provide support to children and adolescents in this situation of migration" (Pelacani and Mantilla 2022).

The social problems faced by migrant children and adolescents involve the whole of society, because while the United Nations Educational, Scientific, and Cultural Organization (Organización de las Naciones Unidas para la Educación, la Ciencia y la Cultura—UNESCO 2021) reflects on the education of the future and sets goals for an inclusive citizenship and a sustainable world by 2050, with a vision of the future based on the analysis of realities such as environmental and climate damage, inequalities, social fragmentation, and political extremism that are leading many societies to a point of crisis, and while these realities are urgent for consideration, it is necessary that the issues surrounding migration do not wait. Indeed, it is difficult to think of a sustainable and harmonious future without these relational, economic, and cultural correctives that need to be addressed based on the policy, rights, especially by the grassroots population, who represent the host localities on a daily basis.

In relation to migrant adolescents, it is urgent to think about their present and their future, to make rules that require an accompanying, more flexible adult and to assume the state as protector and guarantor of their rights regardless of who they travel with, where they arrive, and what their life history is, and to focus on reducing the forms of suffering such as uprooting from their territorial spaces and with them from their representative, imaginary, and symbolic spaces (Espinar Ruiz 2014).

To rethink and act on the serious consequences of the prospective that represents this generation of citizens to whom unequal life has subjected them to school dropout, to a rapid suppression of flexible learning scenarios where they socialize with their peers, and where it was possible to interact with knowledge, values, and a guided understanding of social realities through books, news, or cases, where problem solving was channeled and possible based on mistakes, reflections, and rewriting of other desirable solutions, which is not present in this new life, because there is no possibility of recovering the lost one, one makes the mistake of passing as irregular in a trail, where there are multiple armed actors

and if they want to steal, rape, take away documents, or any situation of demonstration and abuse of power they simply comply (García Pinzón and Trejos 2021).

Therefore, it is essential to think as a society that the human development of the next generations is at stake, their vision of the future, the possibility of building autonomy and with it their moral criteria. A conscience of individual responsibility will allow them a certain quality of life for themselves and their children, for their relationships, for the jobs they will perform as men and women, and for their social role in their territories or in those in which they decide to stay and where it is possible to stay (Andrade 2016).

How will they understand and interact in a social and political citizenship if their memories of that transition from boys to men or girls to women only envision pain, humiliation, exploitation, and low self-esteem in front of states that did not provide them with the defense of their rights just because they did not travel with an adult caregiver or because of the existence of legal loopholes?

These emerging social prospects in south–south migration, which confirm the already mentioned poverty problems demonstrated by the Comisión Económica para América Latina y el Caribe—CEPAL (2019) in which decades of development are set back and where social problems emerge, including migrant children and child labor, is explicitly the only motivation for this migration. In this regard, the Naciones Unidas (2019) declared 2021 the International Year for the Elimination of Child Labor, highlighting the importance of its elimination to close generational poverty gaps that increase inequality and leave no possibility of sustainable development.

It is worth highlighting the damage caused by adolescent labor:

The early involvement of the child or young person in the labor market not only entails sacrifices in terms of their current well-being but also involves a detriment to the well-being expected in the future. All this to the extent that work diminishes the human capital reserves of minors: firstly, because it makes difficult—and sometimes hinders—the time dedicated to education with acceptable quality standards; and, on the other hand, because it increases the risk associated with prolonged exposure to inadequate work environments, which imply excessive and counterproductive effort for the physical and mental development of minors (Acevedo González et al. 2011, p. 114).

This situation leads us to think about the reality of Latin America as a prospective vision of a region that seeks development but does not outline it to the extent that the inequalities that lead to the migration of adolescents are preventing personal, family, and sustainability development after the setbacks due to school dropout, health risks, and elimination of free time and leisure time, which will be shown in social and economic problems that affect circles of poverty, abuse, and social violence.

Finally, the migration of UASA results in an early transition to adult life, compelling them to quickly assume economic responsibility for their survival. This situation often leads to street habitation, child labor, and the rapid acquisition of various skills, particularly related to work. Unfortunately, this accelerated process does not allow them the luxury of experiencing a social moratorium, a crucial period in which they would typically prepare for the maturity and coping skills required for optimal development within their age and life course. This period is essential for strengthening cognitive, social, and emotional development, ultimately setting the stage for a successful transition into adulthood (Urrego Betancourt et al. 2014).

**Author Contributions:** Conceptualization, C.R.-M., N.A.-A., L.M.M.B. and K.G.T.R.; methodology, C.R.-M., N.A.-A., L.M.M.B. and K.G.T.R.; software, C.R.-M. and N.A.-A.; validation, C.R.-M., N.A.-A., L.M.M.B. and K.G.T.R.; formal analysis, C.R.-M., N.A.-A., L.M.M.B. and K.G.T.R.; investigation, C.R.-M., N.A.-A., L.M.M.B. and K.G.T.R.; resources, C.R.-M., N.A.-A.; data curation, C.R.-M. and N.A.-A.; writing—original draft preparation, C.R.-M.; writing—review and editing, C.R.-M., N.A.-A., L.M.M.B. and K.G.T.R.; visualization, C.R.-M., N.A.-A., L.M.M.B. and K.G.T.R. All authors have read and agreed to the published version of the manuscript.

**Funding:** This research was funded by Universidad Simón Bolívar (Colombia) grant number C2060020822.

**Institutional Review Board Statement:** The project from which this study was derived had the ethics review and approval of the institutional ethics committee. The study was carried out in accordance with the Declaration of Helsinki and was approved by the Ethics Committee of the SIMÓN BOLÍVAR UNIVERSITY (COLOMBIA) (In compliance with the Committee's recommendations, the endorsement of the Project CIE-USB-0413-00, was legalized by Act of Project Approval No. 00362 of 22 August 2022) for studies in humans.

**Informed Consent Statement:** Informed consent was obtained from all subjects involved in the study.

**Data Availability Statement:** Data are contained within the references (Ramirez Martinez et al. 2023).

**Conflicts of Interest:** The authors do not report any potential conflict of interest.

## Notes

[1]   Understanding this notion of a separated adolescent traveling with another minor, or with an older person, who cannot guarantee his or her care and protection. Most of them are separated from their primary caretaker relatives.

[2]   This report takes the following age ranges and characteristics: early childhood: 0 to 5 years; infancy: 6 to 11 years; and adolescence: 12 to 17 years.

[3]   The adolescents are put in contact with international foundations that provide them with psychosocial care and protection.

[4]   Means of cargo transportation.

[5]   Illegally hidden among the merchandise being transported.

[6]   Border municipality with Colombia.

[7]   Venezuelan currency that circulated from 2018 to 2021 when five zeros were eliminated, which represents USD 0.0160.

[8]   Border checkpoint guarded by Venezuelan police.

[9]   People who work to help cross the trail, through informal roads and evade migration authorities.

[10]   Group with armed structure, Colombian right-wing extremist group that, despite a demobilization process with the government, operates in a residual manner in different illicit businesses such as control of illegal crossings, smuggling, exchange houses,

[11]   USD 0.63.

[12]   Municipal transportation.

[13]   Non-legal municipal transportation.

[14]   Transported them free of charge.

[15]   Town of Bucarasica, municipality of Norte de Santander.

[16]   USD 14.55.

[17]   USD 2.43

[18]   Pig-raising pen.

[19]   USD 1.21.

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
