# Peer review of "Unaccompanied or Separated Migrant Children and Adolescents at the Colombian–Venezuelan Border: Loss of the Social Moratorium and Its Implications"

_socsci, doi:10.3390/socsci12120683_

Round 1

Reviewer 1 Report

Comments and Suggestions for Authors

Comments for the authors:

This is an important topic and the qualitative research approach is carefully defined and explained. The use of Atlas.ti is helpful in allowing you to identity themes and categorize your participants’ narratives. You might want to refine your initial research questions from “the How and Why” of migration to more specific questions reflecting your focus on the youths’ narratives of their experiences as unaccompanied migrants, their labor experiences, and the dangers they faced which is what your findings focus on. It would be helpful to define where you make the distinction between “child” and “adolescent” as it seems that the majority of your respondents are teenagers, mostly 15-17, although there are a small number of 13–14-year-olds. In the discussion you provide about child and adolescent migrants there are categories of very young children who arrive with a parent or caregiver and numerous ways that children and youth become migrants, but you focus mostly on the older unaccompanied youth/adolescents in your interviews--the group that is more likely to have labor experiences and be exposed to greater dangers on the street. It would be helpful to clarify your focus on the teenage unaccompanied migrants early on in your study.

You provide an excellent overview of the diverse group of irregular migrant children and adolescents and the national and international policies that exist (but are not necessarily implemented) to protect their rights and safety. I was also hoping to see a review of academic studies of the experiences of irregular migrant children and youth focused on Venezuela and Colombia or other Latin American countries. If you could possibly add those to your review of the contexts of irregular youth migration, that would be helpful. For example, Victor Zuniga has done qualitative research on Mexican child migrants when they return to Mexico and Douglas Massey has studied the “culture of migration” among Mexican youth who see migration to the U.S. as a rite of passage because it is so common among their family and community members in key sending areas. Hopefully there are recent studies of youth migration from Colombia, and if not, you could state the need for more such studies as part of your conclusion.

Your main framing concept of social moratorium is discussed well, but it would be helpful to cite other studies of youth migrants that have used this concept if such studies exist. The other concepts such as structural vulnerability, social construction, and interaction are not adequately discussed and perhaps you do not need to refer to them since they do not seem to be key to your analysis in this article. Instead, elaborate more on studies of moral and cognitive development which you emphasize in the conclusion.

The description of the “matrix” designed to organize the category of social moratorium is a bit obtuse and could be explained better. How did you formulate the “guiding questions”? and how did the matrix help you detect “actions” or disruptions of the social moratorium? The four focus points identified on page 7 do not seem to be a part of the findings. There was little discussion of the moment of the decision to migrate, friends who influenced the decisions to migrate, or the actual journey. This article mainly focusses on coping with the contexts of work, violation of rights, and dangers encountered. Perhaps you can clarify the focus of this article vs the broader research themes.

Your methodology section is well done, but I still have some suggestions that I think would be helpful to readers. Define how you composed the “affinity groups” and how and where you conducted the affinity group interviews. What stimulus questions did you ask? Address some of the weaknesses of group interviews, i.e., “group think” when the first respondent sometimes causes others to respond in similar ways, how to assure that everyone participates, how to avoid leading questions, etc. I assumed that those groups were composed of the youth identified in each location, but it was not clear if there were only four groups, how you got them together for the discussions, why there were so few female participants, and whether gender made a difference in participation (i.e., one group has only one male and another only one female). Seems as if gender composition might make a difference when discussing sexual abuse. How did you use the field diaries and who kept them? Who (qualifications/language/age/gender, etc.) conducted the group interviews?

This article is very important in raising the awareness of the national and international policies and laws that exist to protect child/adolescent migrants—even if they do not seem to be enforced. However, the only optimistic area in the article seems to be the youths’ “idealistic but unfair goals” and their “hopeful looks and tones of voice.” They do show determination and a remarkable ability to deal with exploitation by viewing their experiences as “an adventure,”  feeling proud of their courage, leaving exploitive or discriminatory situations when they can (i.e., agency). You refer to their “strength and courage” line 587 and “moral reasoning” line 628, and although you point out that the research [Erikson} considers adolescence as a time of immature thinking and reasoning, perhaps the youths’ “grit” or perseverance and courage to take the initiative to migrate and finding ways to deal with these extreme hardships (friendships, some employment, etc.) will enable some of them to succeed despite the odds. In your conclusion, I would like for you to explore policies or initiatives that perhaps some countries have undertaken to provide innovative and successful resources for these migrants or some non-governmental organizations that have helped or suggested workable policies or interventions to support these youth. You might also suggest other research studies to follow these youth over time to identify strategies for successful outcomes or helpful supports that might be scaled up to policy interventions.

Comments on the Quality of English Language

Overall the English language use is excellent. Some sentences could be shortened for clarity. On line 471 I believe the authors meant minors OVER 17 may work up to 8 hours a day vs. minors UNDER 17. Clarification of the role of the "matrix" versus the tables would help.

Reviewer 2 Report

Comments and Suggestions for Authors

My comments are more about the presentation of the manuscript because I think that substantively, there is little to critique. The authors have an important research question and executed it very flawlessly. I like how they are clear throughout the manuscript about all aspects of the research (the motivation, the contextualization in the literature, the data, the results, etc.). I also think they provide a very interesting narrative about these unaccompanied minors' experiences in Colombia. I think the paper makes a significant contribution and I would support is publication with minor revisions.

The manuscript is very long and there is a lot of material. I like how the authors are covering all aspects of the experience of the minors, but is there any material that could be cut, or moved to an appendix? Also, the presentation of quotes from each participant is very powerful, but I wonder if it would make sense to integrate them into the discussion of the results and move the list format to an appendix? It is a bit overwhelming to see a list of quotes, but that may just be a style preference. The quotes do add a lot of context, and I think they should be kept in the paper. 

I am political scientist, so this is how I approach the paper, but I wonder if there is any context regarding the political situation that is relevant to the experiences that are noted (such as the decision to leave, entering the workforce early, etc.). 

p. 6- line 266- this sentence is missing a word:

"This results in including identity confusion, impulsive behavior, a lack of ma- 266 tured aptitudes and skills, emotional instability, and challenges in forming secure attach- 267 ments and cohesive group relationships"
